# EgoExOR: An Ego-Exo-Centric Operating Room Dataset for Surgical Activity Understanding

**Ege Özsoy**[*]
Technical University of Munich
Munich Center for Machine Learning
`ege.oezsoy@tum.de`

**Arda Mamur**[*]
Technical University of Munich
`arda.mamur@tum.de`

**Felix Tristram**
Technical University of Munich
Munich Center for Machine Learning
`felix.tristram@tum.de`

**Chantal Pellegrini**
Technical University of Munich
Munich Center for Machine Learning
`chantal.pellegrini@tum.de`

**Magdalena Wysocki**
Technical University of Munich
Munich Center for Machine Learning
`magdalena.wysocki@tum.de`

**Benjamin Busam**
Technical University of Munich
Munich Center for Machine Learning
`benjamin.busam@tum.de`

**Nassir Navab**
Technical University of Munich
Munich Center for Machine Learning
`nassir.navab@tum.de`

## Abstract

Operating rooms (ORs) demand precise coordination among surgeons, nurses, and equipment in a fast-paced, occlusion-heavy environment, necessitating advanced perception models to enhance safety and efficiency. Existing datasets either provide partial egocentric views or sparse exocentric multi-view context, but do not explore the comprehensive combination of both. We introduce EgoExOR, the first OR dataset and accompanying benchmark to fuse first-person and third-person perspectives. Spanning 94 minutes (84,553 frames at 15 FPS) of two emulated spine procedures, Ultrasound-Guided Needle Insertion and Minimally Invasive Spine Surgery, EgoExOR integrates egocentric data (RGB, gaze, hand tracking, audio) from wearable glasses, exocentric RGB and depth from RGB-D cameras, and ultrasound imagery. Its detailed scene graph annotations, covering 36 entities and 22 relations (568,235 triplets), enable robust modeling of clinical interactions, supporting tasks like action recognition and human-centric perception. We evaluate the surgical scene graph generation performance of two adapted state-of-the-art models and offer a new baseline that explicitly leverages EgoExOR's multimodal and multi-perspective signals. This new dataset and benchmark set a new foundation for OR perception, offering a rich, multimodal resource for next-generation clinical perception. Our code and data are available at `https://github.com/ardamamur/EgoExOR`.

---

[*]Equal contribution.

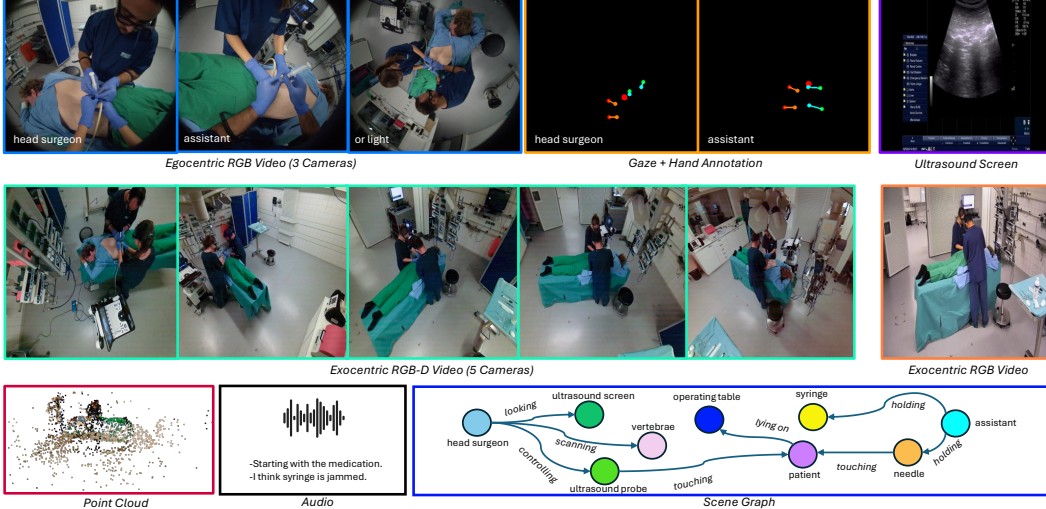

Figure 1: Overview of one timepoint from the EgoExoR dataset, showcasing synchronized multi-view egocentric RGB and exocentric RGB-D video streams, live ultrasound monitor feed, audio, a fused 3D point-cloud reconstruction, and gaze, hand-pose and scene graph annotations. The "closeTo" predicate is not visualized for brevity.

Table 1: Comparison of EgoExOR with existing operating room (OR) datasets. EgoExOR is the first to combine synchronized multi-view egocentric and exocentric recordings with gaze, hand pose, screen capture, and dense scene graph annotations.

| Dataset | Ego. | Multi-View Ego | Multi-View Exo | Gaze | Hand Pose | Screen Recording | Scene Graphs | Annotated Timepoints |
|---|---|---|---|---|---|---|---|---|
| MVOR [1] | | | ✓ | | | | | 732 |
| 4D-OR [2] | | | ✓ | | | | ✓ | 6,743 |
| MM-OR [3] | | | ✓ | | | ✓ | ✓ | 25,277 |
| EgoSurgery [4, 5] | ✓ | | | ✓ | | | | 15,437 |
| **EgoExOR (Ours)** | ✓ | ✓ | ✓ | ✓ | ✓ | ✓ | ✓ | 84,553 |

# 1 Introduction

Modern operating rooms (ORs) are dynamic, safety-critical environments where diverse agents, such as surgeons, nurses, anaesthetists, mobile robots, and the patient, must coordinate seamlessly within a tightly confined space [6, 7]. Every participant has a distinct role, visual perspective, and attentional focus shaped by their responsibilities: a nurse may monitor tool availability, a surgeon may focus on a sub-millimetre needle movement, while an anaesthetist may track patient vitals. Crucially, a scene can evolve in seconds from a calm setup to a crowded, high-stakes intervention. This dynamic development and the multiplicity of perspectives makes the OR a uniquely complex environment, where a momentary lapse or misjudgment can jeopardize patient safety.

Capturing and understanding this complexity requires perception models that move beyond static, ceiling-mounted views and integrate multi-perspective, task-driven viewpoints that reflect the rich structure of human attention and interaction. Perspective-aware and accurate 4D scene understanding, capturing spatial and temporal dynamics of surgical workflows, would enable applications such as automatic documentation, improved team coordination, context-aware robotic or AR guidance, paving the way toward surgical automation. However, enabling such capabilities and making the first steps towards automated surgery is first and foremost **a data problem**. Progress across computer vision has consistently followed the release of large-scale public datasets, such as MNIST [8] and ImageNet [9] in early classification; KITTI [10], Cityscapes [11], and Waymo Open [12] for autonomous driving and Epic-Kitchens [13], Ego4D [14], and Ego-Exo4D [15] for egocentric activity understanding. Surgical Data Science (SDS) has seen similar advances in endoscopic vision, supported by internal

datasets for laparoscopy and arthroscopy [16, 17]. Existing OR datasets, such as MVOR [1], 4D-OR [2], MM-OR [3], and EgoSurgery [4, 5], have advanced scene modeling through multi-view, multimodal, or egocentric data; however, they lack the critical combination of multi-perspective egocentric and exocentric views, gaze, hand pose, and dense scene graphs. This limits holistic OR perception, specifically where multiple agents interact, occlusions are common, and fine-grained manipulations must be inferred from both global context and individual perspectives, hindering applications like context-aware robotic guidance and real-time team coordination.

These limitations highlight the need for a paradigm shift, from relying solely on static, top-down exocentric views to including task-specific perspectives of the OR staff. These perspectives often face the sterile field, circumventing the frequent occlusions in ceiling mounted cameras. Further, each team member has a unique view of the procedure, showing detailed gestures, tools and anatomical landmarks. Modern smart-glasses allow to record sub-millimetre eye-tracking; gaze vectors and hand-tracking, additionally to egocentric views, where each actors' gaze reveals intent and focus and their hand movements reflect granular actions such as grasping a scalpel or manipulating a needle. Together, these elements provide a comprehensive view of surgical activities from each staff's standpoint, essential for developing advanced computer vision models tailored to the OR. Furthermore, as wearable devices do not require modifications to the OR environment, they can be integrated more easily into existing operating rooms.

No existing dataset provides multi-member egocentric perspectives synchronized with exocentric OR views and dense frame-level scene graph annotations. To this end, we introduce EgoExOR, a new dataset and benchmark designed to advance OR scene understanding through an egocentric lens. For the first time, EgoExOR combines multiview egocentric recordings from wearable glasses, including RGB video, audio, gaze, and hand pose data, with multiview exocentric views from RGB-D cameras and screen recordings of ultrasound imaging, as visualized in Figure 1 and described in Table 1. Recorded in a simulated OR environment to ensure ethical and practical feasibility, EgoExOR spans 94 minutes across 41 takes, comprising 84,553 timepoints recorded at 15 FPS. Simulating two key surgical workflows, needle insertion and microsurgery, it delivers synchronized, multimodal data accompanied by scene graph annotations. We establish a new benchmark for surgical scene graph generation, evaluate two state-of-the-art OR scene graph generation models, and introduce a new baseline that leverages all the modalities in EgoExOR.

By uniting the staff members' visual experience with multi-view context and structured annotations, EgoExOR sets the stage for methods that reason jointly about gaze, dexterous manipulation, and the wider clinical scene. Finally, as EgoExOR is the first dataset to combine multiple egocentric and multiple exocentric viewpoints in a synchronized setup and capture simultaneous parallel actions from different agents, we believe it will serve as a cornerstone for the next generation of OR perception models and, more broadly, for any domain where human expertise, attention, and fine motor skill intersect in complex, occluded environments.

## 2   Related Work

**Surgical Data Science.** Surgical Data Science (SDS) has advanced significantly, leveraging deep learning for tasks like action recognition [18, 19], phase identification [20], instrument detection [21], enabled by large datasets like Cholec80 [16] and ArthroPhase [17]. These datasets focus on internal patient views (e.g., laparoscopy, arthroscopy), offering rich annotations but missing the broader OR context, such as interactions among clinical staff or external equipment. Capturing the entire operating theatre is more difficult because cameras must be non-intrusive, patient privacy is paramount and lighting is challenging. The Multi-View Operating Room (MVOR) dataset [1] uses three synchronized RGB-D cameras, capturing 732 frames with coarse pose labels, but its limited scale and coarse annotations restrict its utility. 4D-OR[2] introduced semantic scene graphs for holistic OR modeling, annotating 6,743 timepoints from six ceiling-mounted RGB-D cameras with clinical roles, tools, and interactions. MM-OR [3] scaled this effort, integrating multimodal data and panoptic segmentations for robotic knee surgeries, with an order of magnitude more annotations. However, both rely solely on exocentric views, which do not capture the surgical team members' unique perspectives, are susceptible to occlusions, and lack gaze or hand pose data. These limitations reduce their utility to mainly analysing the top-down perspective on the unobstructed OR room.

**Egocentric Vision.** In the wider field of Computer Vision multiple ego- (and exo-)centric datasets have been proposed to tackle video understanding tasks. Epic-Kitchens [13] for example captures 100 hours of cooking tasks with annotations for actions, objects, and narrations. Epic-Fields [22] extends Epic-Kitchens to include camera poses and sparse SfM pointclouds, enabling object tracking and more comprehensive 3D understanding. HD-EPIC [23] (41h) extends 3D reconstruction annotations and manually curate coarse meshes of the entire scene, which is highly promising for improving the precision of object tracking methods. All the aforementioned datasets, however, focus on cooking and are recorded solely in kitchens, limiting their broader applicability. In contrast, Ego4D [14] was recorded in diverse scenes and settings and spans 3,670 hours, introducing a variety of different benchmarks, from episodic memory over hand-object interaction to action anticipation. For multimodal sensing, Ego-Exo4D [15] combines synchronized egocentric and exocentric views across a combined 1,286 hours (221.26 ego-hours) of skilled activities (e.g., sports, cooking, music), enabling 3D human pose reconstruction and skill assessment. Another multimodal dataset was proposed in HOI4D [24], where a head-mounted RGB-D camera was used to capture egocentric video of human-object interactions. Aria Digital Twin (ADT) [25], recorded with Project Aria glasses, provides 3D reconstructions, audio, and gaze. For procedural tasks, Assembly101 [26] offers 4,321 clips captured from ego and exo perspectives of toy assembly with hand pose annotations.

While these datasets provide important building blocks for general computer vision tasks such as human-object interaction and video understanding in diverse settings, methods developed on them will not generalise to the OR without training data and method design that takes into account the complicated environment of the OR [3]. Except for one work, EgoSurgery [4], there have been no significant efforts to capture egocentric surgical videos, which could provide crucial views of the surgical staffs hands. Specifically, EgoSurgery provides a first-person view from surgeon-mounted camera, with phase and tool labels, capturing actions like suturing but it lacks perspectives from the other staff, exocentric context, team dynamics, or dense scene graphs.

In summary, no single dataset integrates synchronized egocentric and exocentric video, eye tracking, 6-DoF hand-tool trajectories, and dense scene graphs in a clinically realistic OR, which is limiting development of holistic perception methods. EgoExOR addresses this gap with a multimodal, multi-view dataset tailored to precision tasks like needle insertion and microscopic operations, featuring rich annotations that enable reasoning about surgeon attention, intent, action, and clinical context, paving the way for next-generation ego-exocentric OR perception.

## 3  Dataset Acquisition

This section details the acquisition of the EgoExOR dataset, including the recording environment, sensor configurations, participant roles, and emulated surgical procedures, outlining the methodology for acquiring multimodal, multiview data in an OR setting.

### 3.1  Recording Environment

EgoExOR was recorded in an university-affiliated surgical simulation center, previously utilized for the 4D-OR [2] and MM-OR [3] datasets. The center is equipped with surgical tables, anesthesia machines, overhead surgical lights, and standard OR equipment. The layout includes a sterile field with instrument tables and a circulation area for surgical team movement, ensuring a controlled setting for high-fidelity surgical simulations approximating real-world conditions *without* involving real patients. This approach mitigates privacy and safety concerns associated with patient data. While recording data from live surgeries would provide the highest level of realism, such recordings are rarely made publicly available due to strict privacy regulations and ethical considerations [27, 28]. Simulation-based data collection thus remains the most practical approach for creating open, reproducible datasets for surgical scene understanding.

### 3.2  Technical Setup

To capture the rich dynamics of each scenario, we instrumented the environment and participants with a comprehensive set of synchronized sensors, enabling both egocentric (first-person) and exocentric (third-person) views of the action. The following describes the equipment, camera placements, synchronization, and data processing pipeline.

**Egocentric Recording.** We used Project Aria Glasses [29] to capture first-person perspectives from participants (head surgeon, assistant, circulating nurse (circulator), anaesthetist) and non-human viewpoints (surgical microscope for MISS, OR light for Ultrasound). Key specifications were:

- **RGB Cameras**: 1440×1440 resolution at 15 FPS, providing first perspective views of the wearer, essential for resolving subtle actions like needle handling or micro-suture pickup.
- **Eye Tracking Cameras**: 320×240 resolution delivers gaze samples at 120 Hz with sub-millimeter accuracy (2D pixel + depth), enabling precise analysis of surgical intent (e.g., the surgeon's gaze switching between the ultrasound screen and patient).
- **Microphones**: Stereo at 48 kHz, capturing dialogue and ambient sounds.

Wearable cameras were also placed on the microscope (MISS) and OR light to simulate views that would be possible to get from next-generation robotic equipment or OR lights equipped with cameras.

**Exocentric Recording.** To capture the global OR context, we used Azure Kinect Cameras, strategically positioned at the ceiling for full-room coverage. Each records RGB-D images at 15 FPS, offering complementary external views of staff interactions and equipment, and enabling the creation of colored point clouds per timepoint. Camera placements were calibrated using a checkerboard pattern to compute intrinsic and extrinsic parameters, enabling spatial alignment and 3D reconstruction.

**Ultrasound Screen Recordings.** We used an HDMI capture device to record the screen of the ultrasound machine, providing real-time imaging at 15 FPS.

**Synchronization.** Azure Kinect cameras were connected in a master-slave configuration, ensuring frame-level alignment. To ensure all wearable cameras, ultrasound recording, and external recordings were in sync with each other, we used a clapper at the beginning of each take, clearly visible to every camera and audible to all Aria microphones, and manually synchronized the streams. We did not observe any drift in the streams during the duration recordings.

### 3.3 Participants and Roles

EgoExOR features emulated surgical teams composed of biomedical engineers trained to enact specific clinical roles. To enhance procedural variability and mitigate role-specific biases, roles were rotated across recordings, with individuals performing multiple roles and each role executed by different participants. All participants provided written consent for the recording and public release of the dataset, ensuring compliance with ethical standards.

- **Head Surgeon**: Directed the procedures, simulating primary tasks like needle insertion or disc removal.
- **Assistant**: Supported the head surgeon by handing instruments, adjusting equipment, and monitoring the surgical field.
- **Circulator**: Managed the OR environment, prepared tools, and maintained the sterile field.
- **Anaesthetist**: Oversaw the patient's vitals and anesthesia.

### 3.4 Surgical Procedures

EgoExOR models two precision-oriented interventions, **Ultrasound-Guided Needle Insertion (UI)** and **Minimally Invasive Spine Surgery (MISS)**, selected for their prominence in modern spine practice, their representation of distinct imaging-guidance paradigms and their sequential role in treating lumbar disc herniation, a leading cause of lumbar spine surgeries affecting approximately 5 to 20 per 1,000 adults each year [30] and causing radicular pain or lower back pain due to disc pressure on spinal nerves. Lumbar injections such as UI are a widely practiced intervention, with over 1 million lumbar epidural steroid injections administered annually in the United States alone [31]. MISS, particularly lumbar microdiscectomy, is the next step when the injections prove insufficient, removing the herniated disc fragments through a minimally invasive approach, with more than 300,000 procedures performed annually in the U.S. [32]. Although UI and MISS differ in invasiveness and therapeutic scope, they both heavily rely on precise, skillful interaction between the clinicians, tools and patient anatomy in a complex OR environment, making them suitable for studying OR dynamics in a clinically meaningful context. EgoExOR emulated these procedures

following surgical textbooks [33], training videos [34–36], and consultation with clinicians, ensuring alignment with clinical practice guidelines. We defined a phase taxonomy based on standard clinical workflows, segmenting procedures into distinct steps. Recording was organized around this phase structure, and each take corresponds to one or more procedural phases. All takes were scripted and rehearsed in advance to ensure both authenticity and clinical relevance.

**Ultrasound-Guided Needle Insertion (UI).** Simulates an epidural steroid or nerve-root block, where a spinal needle is advanced under real-time ultrasound. Phases:

- **Patient Introduction & Positioning**: The circulator prepares the OR, verbally confirming tool readiness (e.g., "Syringes ready, probe cover in place"), while the head surgeon and assistant position the patient and apply antiseptic.

- **Ultrasound Setup & Target Identification**: The head surgeon adjusts the ultrasound probe to locate landmarks, with the assistant preparing the needle and confirming angles.

- **Medication Injection**: The head surgeon inserts the needle, injects steroids, and observes the spread on the ultrasound screen, while the circulator monitors vitals.

- **Post-Procedure Cleanup**: The needle is removed, a dressing applied, and the team cleans the probe and OR.

**Minimally Invasive Spine Surgery (MISS).** Simulated using a surgical microscope and tubular retractors to remove herniated disc fragments. Phases:

- **OR Preparation & Patient Setup**: The circulator organizes instruments and checks the microscope, while the anaesthetist sets up monitors. The assistant drapes the patient, and the head surgeon confirms positioning.

- **Incision & Initial Access**: The head surgeon makes a small incision, places dilators, and uses the microscope to target the disc, with the assistant monitoring fluoroscopy.

- **Microscope-Guided Discectomy**: The head surgeon removes disc fragments using micro-forceps, confirming decompression, while the assistant ensures a clear field.

- **Wound Closure & Turnover**: The incision is closed with sutures, the patient is woken, and the team sterilizes equipment.

Furthermore, we also designed realistic deviations from typical protocols to reflect the variability observed in surgeries. We scripted complication for each phase, such as dropped instruments (e.g., assistant dropping gauze), needle contamination, ultrasound gel drying, microscope misalignment, syringe jams, or vital-sign fluctuations. Participants responded per standard protocols (e.g., replacing contaminated tools, re-sterilizing gloves), increasing diversity and creating challenging edge cases.

## 4 Dataset Description

This section describes the post-acquisition pipeline that transforms raw multimodal recordings into a structured dataset tailored for OR perception research. We outline the data modalities, processing steps, recording segmentation, statistical overview, and annotation process, providing a comprehensive view of EgoExOR's composition and utility.

### 4.1 Data Processing and Modalities

The EgoExOR dataset comprises multiple synchronized data modalities captured in an OR environment. After acquisition, all data streams undergo a standardized processing pipeline to produce a final, compact dataset suitable for efficient training and evaluation. The following summarizes all included data modalities, their processing and final representations as available in the dataset:

- **Egocentric RGB Video**: Multi-view RGB, down-sampled to $336 \times 336$ pixels, providing first-person perspectives of the OR.

- **Exocentric RGB-D Video**: Multi-view RGB and depth data, downsampled to $336 \times 336$ pixels, offering external perspectives of the OR scene.

Table 2: Distribution of surgical takes, duration, and frames across procedures in EgoExOR.

| Procedure | # Takes | Time [min] | # Frames |
|---|---|---|---|
| Ultrasound-Guided Injection (UI) | 24 | 69 | 62,547 |
| Minimally Invasive Spine Surgery (MISS) | 17 | 25 | 22,006 |
| **Total** | 41 | 94 | 84,553 |

- **Point Cloud**: Depth maps are processed into colored point clouds using calibrated camera extrinsics and reduced to 2,500 points per timepoint.
- **Gaze Tracking**: Eye-tracking data provided as normalized 2D coordinates relative to the egocentric image frame, combined with depth estimates for each gaze sample.
- **Hand Tracking**: Hand pose data from the glasses, with 16 points tracked (8 for the left wrist and palm, 8 for the right wrist and palm).
- **Audio**: Two-channel stereo audio at 48 kHz, normalized to a [-1, 1] range, capturing verbal communication and environmental sounds. Both full audio recordings, aligned with the take, and one-second audio snippets, aligned with individual timepoints are provided.
- **Ultrasound Screen Recordings**: Video captures of ultrasound display, downsampled to 336×336 pixels, enabling synchronized analysis alongside egocentric and exocentric views.

All modalities are temporally synchronized at a uniform rate of 15 FPS, and stored in a consolidated HDF5 archive for easy access. This ensures that visual, auditory, spatial, and other signals are consistently aligned across all timepoints, enabling robust multimodal learning and evaluation.

## 4.2 Dataset Composition and Statistics

EgoExOR consists of 41 takes capturing clinically inspired phases of two emulated surgical procedures: Ultrasound-Guided Injection (UI) and Minimally Invasive Spine Surgery (MISS), as detailed in Table 2. All takes are annotated and synchronized across modalities. The dataset contains a total of 84,553 frames recorded at 15 FPS, spanning 94 minutes. For benchmarking, the dataset is split into training, validation, and test sets at take level, ensuring phase and scenario diversity. Specifically, 26 takes are used for training, 8 takes for validation, and 7 takes for testing. The dataset is 195 GB in size, providing a comprehensive resource for developing and benchmarking OR perception models. A detailed description of the data structure and modalities is provided in the supplementary material.

**Edge-Case Deviations.** In addition to "normal" workflows, EgoExOR systematically integrates edge cases that represent realistic deviations from surgical protocols (e.g., dropped instruments, gel drying, microscope misalignment). Out of the 41 total takes, 15 contain no scripted deviations, while 26 takes include at least one edge case. These 26 takes account for approximately 54k out of 84k total frames (∼64% of the dataset). Importantly, edge cases are restricted to specific sub-phases, and not all frames within such takes depict anomalies. Each edge-case take typically contains multiple instances of the targeted deviation, and role assignments are varied across takes to capture different team dynamics. To support systematic evaluation, we provide a metadata file mapping each take to its surgical sub-phase and indicating whether it includes edge-case events.

## 4.3 Annotations

EgoExOR follows previous works [2, 3], and includes per-frame scene graph annotations to provide structured semantic context, capturing entities (e.g., surgeons, tools, patient) and their relations (e.g., inserting, cutting). Annotations were created manually using a custom interface that allowed annotators to navigate frames, zoom into details, and overlay multimodal data (e.g., gaze points on RGB frames). For each frame, one trained annotator labeled entities and relations, with a second annotator verifying the output to ensure accuracy.

The annotation schema comprises **36 entity classes** (e.g., `head_surgeon`, `scalpel`, `ultrasound_probe`, `patient`) and **22 relation classes** (e.g., `holding`, `using`, `closeTo`), reflecting clinical roles, tools, anatomical landmarks, and their interactions. On average, each frame contains **6.8 ± 2.5 relation triplets** (median = 7, max = 13), with a total of 568,235 triplets across all 84,553 frames. In the appendix we provide exact statistics for the distribution of all classes.

# 5   Scene Graph Generation Benchmark

This section presents our benchmark for surgical scene graph generation, trained and evaluated on the EgoExOR dataset. All experiments were conducted on a single NVIDIA A40 GPU (48GB) using PyTorch 2.0.1, CUDA 11.7, and Python 3.11, and training time was approximately 5 days.

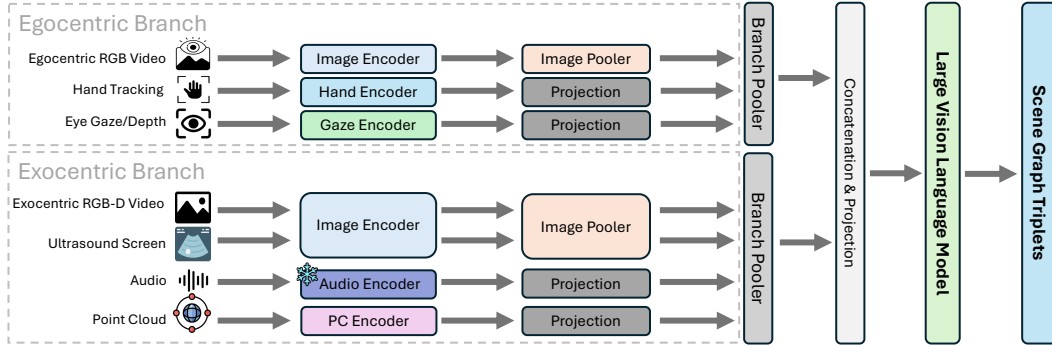

Figure 2: Overview of the proposed EgoExOR model for surgical scene graph generation. The model employs a dual-branch architecture to separately process egocentric and exocentric modalities. Fused embeddings are passed to a large language model (LLM) to autoregressively generate scene graph triplets representing entities and their interactions.

The surgical scene graph generation task [37, 2, 38, 39] aims at generating structured graph representations of OR scenes with nodes as entities (e.g., surgeon, patient, tools) and edges as interactions (e.g., *holding*, *lying on*), summarizing the semantics of the scene. Following prior work [40, 3], we represent scene graphs as triplets (*subject, predicate, object*), capturing dynamic interactions like *(assistant, aspirating, patient)* and spatial relationships like *(patient, lying on, operating table)*. We train and evaluate two baseline models, ORacle [40] and MM2SG [3], adapted to process EgoExOR's ego-exo images. ORacle employs a 2D multi-view visual encoder to embed OR scenes, where as MM2SG extends this incorporating additional modalities such as point clouds, ultrasound screen recordings, and audio. Both use a large language model (LLM) to autoregressively predict scene graph triplets. Both process egocentric and exocentric RGB inputs jointly within a single shared encoder, and lack dedicated fusion mechanisms for perspective-specific features, and are unable to leverage EgoExOR's hand and gaze tracking signals.

**EgoExOR Model.** To fully exploit EgoExOR's rich multi-perspective data, we introduce a new baseline model featuring a dual-branch architecture Figure 2. The *egocentric branch* processes first-person RGB, hand pose, and gaze data, while the *exocentric branch* handles third-person RGB-D, ultrasound recordings, audio, and point clouds. Each branch uses a 2-layer transformer to fuse its inputs into $N$ feature embeddings. These are concatenated and fed into the LLM for triplet prediction. By explicitly separating and fusing perspective-specific features, our model better captures actions and staff interactions, outperforming single-stream baselines in modeling complex OR dynamics.

**Evaluation Metric.** Following established protocols [2, 40, 3], we evaluate performance using the macro F1-score over predicates, assigning equal weight to each class to account for class imbalance.

**Implementation Details.** All models use LLaVA-7B [41] as the starting point, with Vicuna-7B [42] as the LLM and CLIP ViT [43] as the image encoder. We fine-tune using LoRA [44] for the LLM and adapt the last 12 layers of the image encoder for the OR domain, following MM2SG [3]. The number of fixed image tokens $N$ is set to 576. Audio is encoded with CLAP [45], and point clouds with Point Transformer V3 [46]. For the EgoExOR model, both hand pose data and gaze/gaze-depth data are encoded using MLPs, and then projected into a shared latent space as single tokens. All models were trained for 4 epochs. The adapted ORacle and MM2SG baselines also process EgoExOR's egocentric RGB alongside exocentric data, however in a unified stream, lacking the EgoExOR model's specialized branch for egocentric signals.

**Results.** Overall the results, shown in  Table 3 and Figure 3, demonstrate that more modalities lead to improved results, and while the existing works perform satisfactorily, the dual-branch EgoExOR model achieves the highest macro F1.  Several predicates such as *injecting, aspirating, holding,*

Table 3: Scene graph generation results on EgoExOR. The table reports macro F1 scores for two surgical procedures: Ultrasound-Guided Injection (UI) and Minimally Invasive Spine Surgery (MISS). Used modalities such as egocentric and exocentric RGB images (Images), ultrasound screen (Ultra.), audio, point cloud (PC), gaze and hand pose (Hand) are indicated with a checkmark.

| Model | Images | Ultra. | Audio | PC | Gaze | Hand | UI | MISS | Overall |
|-------|--------|--------|-------|----|----|------|----|------|---------|
| ORacle [40] | ✓ | | | | | | 0.65 | 0.66 | 0.63 |
| MM2SG [3] | ✓ | ✓ | ✓ | ✓ | | | 0.72 | 0.66 | 0.67 |
| **EgoExOR (Ours)** | ✓ | ✓ | ✓ | ✓ | ✓ | ✓ | **0.79** | **0.68** | **0.72** |

*controlling* in EgoExOR rely on understanding transient tool-hand trajectories, and fine-grained action cues. This emphasizes the importance of explicitly modeling multiple viewpoints and leveraging all available modalities to improve OR scene understanding. However, it still struggles with low-frequency predicates such as *cutting, positioning*. We provide detailed per-predicate performance metrics and further visualizations in the supplementary material.

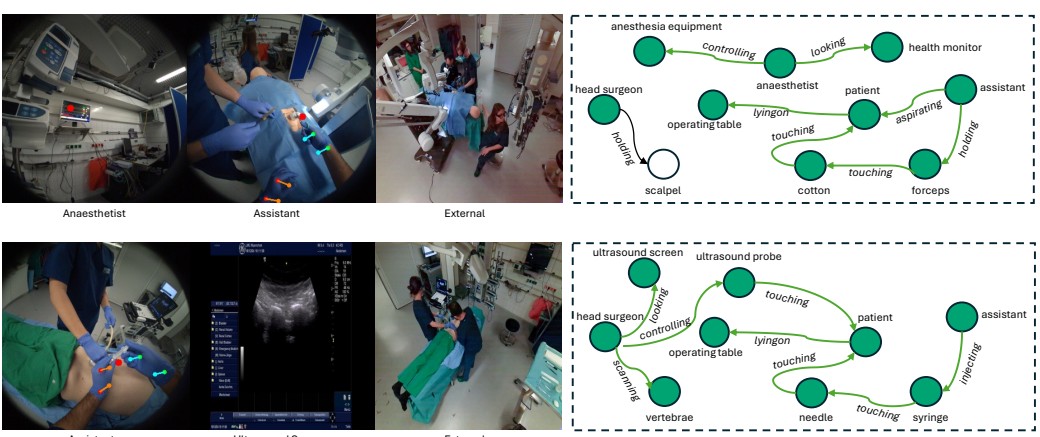

Figure 3: Qualitative examples from EgoExOR. Correctly predicted entities and predicates are highlighted in green, wrong ones are left white. The "closeTo" predicate is not visualized for brevity.

## 6 Conclusion

EgoExOR marks a significant advance in surgical data science, delivering a comprehensive dataset that captures the intricate dynamics of operating rooms through synchronized egocentric and exocentric viewpoints. Across 84,553 frames of two simulated spine procedure, it combines diverse modalities with rich scene graph annotations to model clinical workflows holistically. Our benchmark, comparing existing OR perception models against a new method designed for EgoExOR, underscores its value for developing intelligent systems, from automated documentation to context-aware robotic assistance. EgoExOR's fine-grained action detection, presents unique challenges, making it a valuable resource for advancing capabilities essential for advancing OR perception and human-centric AI in complex, safety-critical environments. We believe EgoExOR will catalyze innovation in OR perception, paving the way for smarter, safer surgical environments, with implications for any field requiring precise human-robot collaboration in complex environments.

**Ethical Considerations.** By employing simulated procedures, EgoExOR avoids privacy concerns inherent to clinical data, enabling unrestricted public release and reproducibility in alignment with NeurIPS ethical standards. All participants provided informed consent for data collection and publication. Potential positive societal impacts include accelerating the development of intelligent clinical assistance systems that improve surgical safety, support clinical decision-making, and enhance training through rich multimodal datasets. As with any technology in sensitive domains, there is also a potential negative impact if such models are deployed prematurely or without sufficient human oversight, potentially leading to over-reliance on automated systems in high-stakes clinical

environments. We emphasize that EgoExOR is intended as a research resource, not as a clinical decision-making system.

**Limitations.** While the simulated procedures in EgoExOR provide a valuable ressource for multi-perspective and multi-sensor OR understanding, its simulated setup can not fully reflect the intricacies of expert surgical performance in live clinical settings. Additionally, the recordings are limited to a specifically equipped operating room that enabled high-quality, synchronized multimodal data but may restrict the adaptability of models to varied OR configurations. Future work could expand this scope, building on EgoExOR's robust foundation.

**Acknowledgements.** We thank INM and Frieder Pankratz in helping setting up the acquisition environment, and Felix Holm and Miruna-Alexandra Gafencu for their help in data acquisition. Authors would like to thank Carl Zeiss AG for their partial support. We also thank Carl Zeiss AG for their partial financial support, and gratefully acknowledge Meta Reality Labs for providing Aria research glasses.

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
