# OpenReview forum: "EgoExOR: An Ego-Exo-Centric  Operating Room Dataset for Surgical Activity Understanding"
_NeurIPS.cc/2025/Datasets_and_Benchmarks_Track — NeurIPS 2025 Datasets and Benchmarks Track poster_

### Official Review · Reviewer_kA8o · 2025-06-14

**Rating:** 5
**Confidence:** 4

**Summary:**

1. First-of-its-kind dataset merging first- and third-person perspectives in the OR: EgoExOR includes egocentric data (RGB video, gaze, hand tracking, audio) from wearable glasses, as well as exocentric RGB and depth data from RGB-D cameras and ultrasound imagery.
2. Detailed scene graph annotations: Covering 36 entities and 22 relations, enabling robust modeling of clinical interactions and supporting tasks like action recognition and human-centric perception.
3. New benchmark for surgical scene graph generation: Establishing a new standard by evaluating two state-of-the-art models and introducing a new baseline that leverages all modalities in EgoExOR.
4. Advancing OR perception: Setting the stage for methods that reason jointly about gaze, dexterous manipulation, and the wider clinical scene, pushing the boundaries of OR perception.

**Dataset Code Accessibility:**

Yes

**Ethical Considerations:**

No, there are no or only very minor ethics concerns

**Final Justification:**

The authors have addressed my concerns.

**Limitations Weaknesses:**

1. Simulation-based data: While the use of simulations allows for ethical and practical data collection, it may not fully capture the complexities and nuances of real-world surgical procedures.
2. Limited generalizability: The dataset is recorded in a specific OR setup, which may limit the generalizability of models to different environments and equipment configurations.
3. Lack of real patient data: The absence of real patient data means that the dataset may not fully capture the emotional and psychological aspects of surgery, which could be relevant for certain applications.

**Strengths Contributions:**

1. Novel and comprehensive approach: EgoExOR is the first dataset to combine egocentric and exocentric perspectives in a surgical context, providing a rich and diverse dataset for OR scene understanding.
2. Multi-modal data collection: The dataset includes various modalities such as RGB video, depth data, gaze, hand pose, and audio, enabling the development of robust and versatile models.
3. Detailed annotations: The inclusion of scene graph annotations with 36 entities and 22 relations allows for precise modeling of clinical interactions and supports complex tasks.
4. New benchmark and baselines: EgoExOR provides a new benchmark for surgical scene graph generation and introduces improved baselines that leverage the unique features of the dataset.

---

> ### Author Rebuttal · Authors · 2025-07-30
>
> We thank the reviewer for their constructive feedback and thoughtful assessment. We appreciate your recognition of EgoExOR as the **“First-of-its-kind dataset merging first- and third-person perspectives in the OR”** and for highlighting its **“multi-modal data collection”** and **“detailed scene graph annotations… covering 36 entities and 22 relations.”** We're also grateful that you view our dataset and baselines as “setting the stage for methods that reason jointly about gaze, dexterous manipulation, and the wider clinical scene.” We address your comments on simulation and generalizability in detail below.
>
> **\[1\] Simulation-based data.**
>
> We appreciate the reviewer’s comments on the limitations of simulation-based data. Our dataset was recorded in a university-affiliated surgical simulation center, and equipped with **real OR infrastructure, including surgical tables, surgical lights, an anesthesia machine, ultrasound device and surgical microscope.**
>
> While we fully agree that live surgical recordings would offer the highest realism, as highlighted in our manuscript, **such data are rarely made publicly available due to stringent privacy regulations and ethical considerations \[27, 28\]**. In contrast, our simulation-based approach allows us to provide a rich, multimodal, reproducible and publicly available dataset that approximates real-world surgical workflows while adhering to privacy and ethical standards.
>
> **\[2\] Generalizability: The dataset is recorded in a specific OR setup, which may limit the generalizability of models to different environments and equipment configurations.**
>
> We agree and acknowledge that the single operating room layout can constrain the dataset’s direct generalizability. However, we believe this reflects the practical challenges of collecting and publicly releasing synchronized multimodal surgical data at scale. Capturing and releasing egocentric and exocentric video, gaze, audio, hand pose, ultrasound screens, and 3D spatial information in a real OR is logistically and ethically difficult. EgoExOR represents a first step in this direction and establishes a new benchmark for surgical scene understanding. We hope it encourages the community to further expand data collection efforts to cover additional procedures and diverse clinical sites.

---

> > ### Comment · Reviewer_kA8o · 2025-08-07
> > **Thanks for you rebuttal, you have addressed my concerns**
> >
> > Thanks for you rebuttal, you have addressed my concerns

---

> > > ### Author Response · Authors · 2025-08-07
> > >
> > > Thank you again for your feedback and for confirming that our rebuttal addressed your concerns. If the clarifications strengthened your view of the work, we’d be grateful if you’d consider updating your score to reflect that.

---

> ### Comment · Area_Chair_5CTf · 2025-08-04
>
> Dear kA8o,
>
> Please make sure to read the other reviews and the author response and engage in an open exchange with the authors asap so there is time for back and forth discussion.
>
> Best, AC

---

> ### Comment · Area_Chair_5CTf · 2025-08-06
>
> Dear kA8o,
>
> Please read the other reviews and the author response asap and let me and the authors know if you have any remaining concerns.
>
> Particularly, were your concerns about the simulations, limited generalizability and lack of real patient data addressed by the rebuttal?
>
> Best, AC

---

### Official Review · Reviewer_LrH6 · 2025-07-01

**Rating:** 5
**Confidence:** 4

**Summary:**

This paper presents EgoExOR, a novel multimodal dataset capturing simulated operating room (OR) scenes to advance research in surgical scene understanding and perception. It includes first-person videos, gaze, hand tracking, and audio captured from wearable glasses, as well as third-person video, depth, and ultrasound images obtained under two types of simulated surgeries. The dataset contains 41 takes with detailed scene graph annotations describing clinical roles, tools, and interactions. The authors evaluate the surgical scene graph generation performance of two adapted state-of-the-art models and introduce a new dual-branch baseline that separately processes first-person and third-person data.

**Dataset Code Accessibility:**

Yes

**Ethical Considerations:**

No, there are no or only very minor ethics concerns

**Final Justification:**

The authors give an insightful and detailed response. I appreciate the clarifications provided and have updated my score accordingly.

**Limitations Weaknesses:**

1. The dataset includes realistic deviations from standard surgical protocols (e.g., dropped instruments, gel drying, etc.) to simulate edge cases. However, the paper does not quantify how frequently these complications occur or how they are distributed across the dataset. Could the authors provide statistics on the prevalence of such cases and clarify whether they are annotated or separately flagged? Additionally, how do these edge cases influence model performance—are they used in evaluation or merely present as background variation?
2. The paper only compares two baselines (ORacle and MM2SG). Given the growing interest in multimodal scene understanding, incorporating a wider range of baselines would enhance the strength of the benchmark. For example, the following recent works could be relevant: 4D Panoptic Scene Graph Generation NeurIPS 2023 \\ HIG: Hierarchical Interlacement Graph Approach to Scene Graph Generation in Video Understanding CVPR 2025 \\ and Learning 4D Panoptic Scene Graph Generation from Rich 2D Visual Scene CVPR 2025 \\. Although these methods were not originally designed for surgical scenarios, they could potentially be applied to this domain and offer meaningful comparisons.
3. While the dual-branch model design is compelling, the paper does not include an ablation study to quantify the contribution of each modality (e.g., gaze, hand pose, ultrasound, audio). Such analysis would provide deeper insight into which components drive performance gains.

**Strengths Contributions:**

1. EgoExOR includes multimodal first-person and third-person data recorded in a simulated operating room, addressing the limitations of existing datasets in terms of surgical perspectives and modalities.
2. Detailed scene graph annotations are provided for each video frame, covering entities, tools, and interactions, combined with multimodal information such as gaze and hand movements, offering a solid foundation for studying complex surgical behaviors and team interactions.
3. The paper establishes an evaluation benchmark on the dataset and proposes a novel dual-branch model that effectively fuses egocentric and exocentric modalities, demonstrating superior performance compared to previous baselines.

---

> ### Author Rebuttal · Authors · 2025-07-30
>
> We thank the reviewer for their thoughtful and constructive feedback. We appreciate your recognition that EgoExOR **“addresses the limitations of existing datasets in terms of surgical perspectives and modalities”**, and that it provides **“detailed scene graph annotations… combined with multimodal information such as gaze and hand movements.”** We’re also grateful that you found our dual-branch model to be a strong contribution, “demonstrating superior performance compared to previous baselines.” We address your comments and suggestions below.
>
> **\[1\] While the dual-branch model design is compelling, the paper does not include an ablation study to quantify the contribution of each modality (e.g., gaze, hand pose, ultrasound, audio). Such analysis would provide deeper insight into which components drive performance gains.**
>
> We thank the reviewer for this suggestion. While our main focus was on building and evaluating a strong multimodal benchmark and model, we do provide some analysis of modality effect. In Table 3 of the main paper, we compare ORacle, which only uses images, to MM2SG which additionally uses ultrasound, pointclouds, and audio, to our baseline model, which uses all the modalities, showing performance gains from using richer multimodal information. Furthermore, Table 4 in the supplementary material provides an **ablation between egocentric-only and exocentric-only inputs, showing that egocentric streams yield considerably higher performance**. While we agree that a more detailed modality ablation, such as training our baseline model with one modality added at a time, could provide further insights, due to the limited rebuttal timeline and high compute cost of training and evaluating multiple models (each requiring ~5 days), we were unable to perform these additional experiments for the rebuttal.
>
> **\[2\] The paper only compares two baselines (ORacle and MM2SG). Given the growing interest in multimodal scene understanding, incorporating a wider range of baselines would enhance the strength of the benchmark. For example, the following recent works could be relevant….**
>
> We appreciate the reviewer’s reference to recent strong baselines. However, most of these models, such as PSG or 4D Panoptic Scene Graph Generation, are designed for **single image** or **single point cloud input**, and therefore **do not natively support synchronized multi-view or multimodal inputs**. As shown in MM-OR paper [3], both **ORacle and MM2SG significantly outperform a naive adaptation of PSG to multiview data in surgical domain**, achieving F1 score improvements of 14.4% and 20.1%, respectively.
>
> For these reasons, we consider it sufficient to compare EgoExOR against these two strong, domain-relevant multimodal baselines. We will clarify this reasoning in the revised version and hope that the introduction of EgoExOR helps catalyze the development of future scene graph approaches that can better handle the unique challenges of surgical data.
>
> **\[3\] The dataset includes realistic deviations from standard surgical protocols (e.g., dropped instruments, gel drying, etc.) to simulate edge cases. However, the paper does not quantify how frequently these complications occur or how they are distributed across the dataset.**
>
> Thank you for highlighting this important point. The recordings are organized into takes, where **a take can either simulate a “normal” or “edge case” workflow**, where specific deviations from standard protocol (e.g., dropped instruments, gel drying, contamination) are intentionally introduced. Each edge-case take typically contains multiple instances of the targeted anomaly, with further slight variations between them. Additionally, we also vary the role assignments across takes, ensuring that different team members perform diverse tasks. We currently use these edge-case both for training and evaluation, our dataset makes it possible to systematically study model robustness to rare but clinically significant deviations. We believe this opens new research directions and invites the community to leverage this resource for developing and evaluating more robust and generalizable models.
>
> Out of the **42 total takes** in EgoExOR, **15 contain no scripted edge-case deviations**, while the remaining **27 takes include at least one such event**. These 27 takes, account for \~**54k out of \~84k** frames (\~64% of the dataset). However, the presence of an edge case in a take does **not** imply that all frames in that take depict deviations, edge cases occur only within specific sub-phases. We will provide this information, as well as the following table in the supplementary material detailing the **edge-case types**, the **phases in which they occur**, and **how often each is present**.
>
> | Procedure | Phase | Sub-Phases | Scripted Edge-Case Deviations |
> | --------- | ---------------------------- | --------------------------- | --------------------------------------------------------------------------------------------------------------------------- |
> | UI | Patient intro & positioning | OR preparation, patient entry, skin prep | Missing instrument; patient reposition; dropped tool |
> | | Ultrasound setup & targeting | Ultrasound guidance | Gel dry-out; body-marker used when ultrasound landmarks unclear; re-target area due to unexpected change in target location |
> | | Medication injection | Needle insertion, steroid injection | Syringe jam; needle reposition |
> | | Post-procedure cleanup | Needle removal, dressing, cleanup | — |
> | MISS | OR prep & patient setup | Room preparation, disinfection | Missing instrument; drape obscure; dropped tool |
> | | Incision & initial access | Incision, target identification | Microscope misalignment; patient movement due to unexpected response requiring anesthesia adjustment |
> | | Microscope-guided discectomy | Disc fragment removal | Scalpel needed again due to unexpected herniated disc morphology; aspiration needed |
> | | Wound closure & turnover | Closure, patient wake-up, cleanup | — |
>
>
> Finally, to clarify the dataset structure and enable filtering, **we will add a metadata file to the dataset itself, indicating the mapping between each take, its corresponding surgical subphase (e.g. “Microscope-Guided Discectomy”), along with a flag denoting whether it includes an edge case**. This will allow users to isolate or analyze specific scenarios more easily.

---

> > ### Comment · Reviewer_VS4V · 2025-08-08
> >
> > I would like to thank the authors for the rebuttal. Majority of my concerns are addressed, personally, really like this dataset.
> > After reading all the reviews and the rebuttal, I would like to keep my original rating.

---

> > > ### Author Response · Authors · 2025-08-08
> > >
> > > Thank you again for your constructive review and for noting that most of your concerns were addressed. We’re especially glad to hear you really like the dataset and appreciate your engagement throughout the process.

---

### Official Review · Reviewer_3wHi · 2025-07-01

**Ethics Flags:** Data privacy, copyright, and consent
**Rating:** 4
**Confidence:** 3

**Summary:**

This paper presents EgoExOR, a novel dataset designed for surgical scene understanding in operating rooms (ORs). It integrates synchronized egocentric and exocentric multimodal recordings (RGB, depth, gaze, audio, ultrasound screen, hand pose, point cloud) captured during two simulated spine surgery procedures: Ultrasound-Guided Needle Insertion (UI) and Minimally Invasive Spine Surgery (MISS). The dataset comprises 84,553 frames, each annotated with detailed scene graphs (568,235 triplets across 36 entity and 22 relation classes).
A new dual-branch model is proposed to leverage this dataset for surgical scene graph generation, outperforming two baselines (ORacle and MM2SG).

**Additional Feedback:**

1.	This point is more for discussion than criticism. While the dataset covers a rare and important setting, I have some doubts about its practical utility. Specifically, how useful is the annotation of entity and relation classes and the task of scene graph generation itself for real-world applications in the operating room? Are these annotations and tasks too simple?

**Dataset Code Accessibility:**

Yes

**Ethical Comments:**

No.

The dataset was collected in a simulated surgical environment using an emulated surgical team composed of biomedical engineers who were trained to perform specific clinical roles. The only potential ethical consideration relates to the consent of these participants, which the authors have already addressed. No other privacy, safety, or misuse risks are identified.

**Ethical Considerations:**

Yes, there are ethics concerns that require attention by the authors

**Limitations Weaknesses:**

1.	There is a factual inaccuracy in the motivation. In the fourth paragraph of the Introduction, the authors state: “No dataset bridges the gap between OR-specific needs and egocentric vision…” However, the EgoSurgery dataset already includes egocentric vision. The novelty here lies in multi-view egocentric data from different team members, which should be clarified.
2.	The claim about synchronization is questionable. The authors rely on a clapper to synchronize wearable cameras and ultrasound recordings, but since the cameras record at only 15 FPS, any asynchronous start may lead to noticeable drift.
3.	The experiments are somewhat limited. While not essential, additional ablation studies exploring the contribution of each modality to scene graph generation would strengthen the paper.
4.	Figure 3 lacks proper referencing and analysis in the main text.

**Strengths Contributions:**

1.	This paper introduces a multimodal surgical dataset that addresses a highly data-scarce domain and is valuable for advancing next-generation OR perception models.
2.	A key highlight of the dataset is the inclusion of multi-member egocentric views, which are crucial for modeling surgical interactions but overlooked in prior datasets.
3.	The data collection process and dataset structure are clearly and thoroughly described.

---

> ### Author Rebuttal · Authors · 2025-07-30
>
> We thank the reviewer for appreciating and highlighting our dataset as **“unmistakably novel”**, its **high data quality** (“Visual inspection shows quality on par with benchmarks like EgoExo4D, EPIC-Kitchens, and EgoHumans”), our baseline model and benchmark, as well as the field of surgical scene understanding as “important”. We appreciate your feedback and have addressed your comments and questions below.
>
> **\[1\] While not essential, additional ablation studies exploring the contribution of each modality to scene graph generation would strengthen the paper.**
>
> We thank the reviewer for this suggestion. While our main focus was on building and evaluating a strong multimodal benchmark and model, we do provide some analysis of modality effects. In Table 3 of the main paper, we compare ORacle, which only uses images, to MM2SG which additionally uses ultrasound, pointclouds, and audio, to our baseline model, which uses all the modalities, showing performance gains from using richer multimodal information. Furthermore, Table 4 in the supplementary material provides an **ablation between egocentric-only and exocentric-only inputs, showing that egocentric streams yield considerably higher performance**. While we agree that a more detailed modality ablation, such as training our baseline model with one modality added at a time, could provide further insights, due to the limited rebuttal timeline and high compute cost of training and evaluating multiple models (each requiring \~5 days), we were unable to perform these additional experiments for the rebuttal.
>
> **\[2\] This point is more for discussion than criticism. While the dataset covers a rare and important setting, I have some doubts about its practical utility. Specifically, how useful is the annotation of entity and relation classes and the task of scene graph generation itself for real-world applications in the operating room? Are these annotations and tasks too simple?**
>
> We agree that highlighting the potential applications of our dataset and scene graphs would further increase the adoption of our dataset. Our scene graph annotations offer a **structured and semantically rich representation of entities and interactions in the surgical environment**. They capture all active entities and their interactions at a given timepoint. **This format aligns well with existing work in surgical scene understanding and enables downstream reasoning tasks** such as **phase recognition** and **role prediction** (Holistic OR domain modeling: a semantic scene graph approach by Özsoy et al., International Journal of Computer Assisted Radiology and Surgery 2024), **sterility breach detection** [3], **next step prediction** (A Group Activity Based Method for Early Recognition of Surgical Processes Using the Camera Observing Surgeries in an Operating Room and Spatio-Temporal Graph Based Deep Learning Model by Nishikawa et al., 2025) and **video synthesis** (VISAGE: Video Synthesis Using Action Graphs for Surgery by Yeganeh et al., MICCAI 2024.
>
> Furthermore, from a benchmark perspective, the scene graph generation task is still not saturated. Our best-performing model currently achieves 0.79 F1 overall, and worse on some predicates, such as “anaesthetising” or “cutting”, at 0.02 and 0.20 F1 respectively, well below human-level understanding. This highlights the difficulty and the continued research value of the task.
>
> We will discuss and emphasize these and other potential use cases in the final manuscript.
>
> **\[3\] The authors rely on a clapper to synchronize wearable cameras and ultrasound recordings, but since the cameras record at only 15 FPS, any asynchronous start may lead to noticeable drift.**
> We appreciate the reviewer’s concern regarding synchronization. We clarify that synchronization was performed **manually** based on the clapper event at the start of each session and was then **visually verified at the end of each recording to ensure that no noticeable drift occurred**. While minor temporal misalignment is theoretically possible due to the 15 FPS frame rate (i.e., one frame every 66.7 ms), the maximum uncertainty in aligning any visual event is half a frame, or approximately ±33.3 ms. In practice, **we did not observe any perceptible or distracting misalignment in our quality checks**. We will clarify this process in the revised version of the paper.
>
> **\[4\] In the fourth paragraph of the Introduction, the authors state: “No dataset bridges the gap between OR-specific needs and egocentric vision…” However, the EgoSurgery dataset already includes egocentric vision. The novelty here lies in multi-view egocentric data from different team members, which should be clarified.**
>
> We agree and will revise the sentence in the final manuscript to accurately acknowledge EgoSurgery’s egocentric stream while clarifying that our novelty lies in multi-member egocentric and exocentric synchronization and dense frame-level scene graphs.
>
> We will also ensure that Figure 3 is properly referenced and analyzed in the text.

---

> > ### Comment · Reviewer_3wHi · 2025-08-06
> >
> > Thanks for the authors’ response. Most of my concerns have been adequately addressed. However, regarding the “practical utility,” I was hoping for more than a speculative discussion, ideally, some form of experimental verification in these application scenarios. I encourage the authors to explore such applications in future work. Given this, I will maintain my current score.

---

> > > ### Author Response · Authors · 2025-08-06
> > >
> > > Thank you for your thoughtful feedback and for noting that our responses addressed most of your concerns. While our current submission focused on establishing EgoExOR dataset and its baseline model, we appreciate your suggestions on exploring diverse application scenarios. We see this as a valuable direction for future work, both by us and the broader community, to further highlight its real-world impact.

---

> ### Comment · Area_Chair_5CTf · 2025-08-04
>
> Dear 3wHi,
>
> Please make sure to read the other reviews and the author response and engage in an open exchange with the authors asap so there is time for back and forth discussion.
>
> Best,
> AC

---

> ### Comment · Area_Chair_5CTf · 2025-08-06
>
> Dear 3wHi,
>
> Please read the other reviews and the author response asap and let me and the authors know if you have any remaining concerns.
>
> Particularly, were your concerns about the motivation, synchronization and limited experiments addressed by the rebuttal?
>
> Best, AC

---

### Official Review · Reviewer_VS4V · 2025-07-09

**Rating:** 5
**Confidence:** 4

**Summary:**

- The paper proposes a multiview video dataset for surgical activity understanding.
- Views: Each participant’s egocentric perspective is captured with Aria glasses, while static third-person RGB-D cameras record the exocentric scene. All cameras are time-synchronized, and the dataset also includes audio, gaze, and hand-tracking streams.
- Dataset statistics: 94 minutes (≈84 600 frames) at 15 fps covering two simulated spine procedures.
- Focus task: Action recognition via surgical scene-graph prediction, which models interactions among entities and their relations.
- Evaluation: A multimodal model trained on the dataset reliably interprets the surgical environment.

**Dataset Code Accessibility:**

Yes

**Ethical Considerations:**

No, there are no or only very minor ethics concerns

**Final Justification:**

This dataset is a great contribution and is presented well.
All of my concerns were addressed, after reading the other reviews and the rebuttal, rating the work as accept.

**Limitations Weaknesses:**

- Choice of task: Scene-graph generation is a sensible entry point, yet the current shift toward natural-language video understanding suggests that relying solely on sparse, explicit graphs may limit the benchmark’s impact. Expanding the suite to include tasks such as dense captioning, action recognition, object-of-interest detection, and segmentation would attract a broader user base.

- Narrow scope: The dataset features only two simulated spine procedures—ultrasound-guided injection and micro-discectomy—spanning 94 min (~84 k frames) in a single OR layout. This limited diversity raises questions about generalization to other surgical environments.

**Strengths Contributions:**

- Presentation: The paper is well written, organized, and easy to follow.

- Importance of the problem: The dataset targets a critical facet of surgical understanding in operating rooms. Multimodal resources like this are foundational for advancing research toward teleoperated or autonomous surgical systems that reduce human error.

- Novel dataset: Compared with existing collections such as MVOR, 4D-OR, and EgoSurgery, the proposed dataset is unmistakably novel, as highlighted in Table 1. It also offers an order of magnitude more data and annotations than previous resources.

- Quality of data: Visual inspection shows quality on par with benchmarks like EgoExo4D, EPIC-Kitchens, and EgoHumans. Annotations are dense, pairing scene point clouds with detailed scene graphs that cover 36 entities and 22 relations.

- Baseline model: The paper establishes a benchmark for scene-graph generation. Inputs include egocentric video, hand tracking, eye gaze, exocentric video, ultrasound screen footage, audio, and point clouds. Each modality is encoded and fused with a pretrained vision–language model, yielding promising F1 scores on the test set.

---

> ### Author Rebuttal · Authors · 2025-07-30
>
> We thank the reviewer for their detailed and constructive feedback. We are grateful for your recognition that our work “**introduces a multimodal surgical dataset** that addresses a highly data-scarce domain and is **valuable for advancing next-generation OR perception models**.” We also appreciate your highlighting of the multi-member egocentric views as “a key highlight of the dataset… crucial for modeling surgical interactions but overlooked in prior datasets.” Finally, we thank you for noting that “the data collection process and dataset structure are clearly and thoroughly described.” We address each of your comments and suggestions in detail below.
>
> **\[1\] Task Selection Rationale:  Scene-graph generation is a sensible entry point, shift towards natural-language, dense captioning…**
>
> We appreciate the reviewer’s suggestion and fully agree that additional annotations such as dense captions, action labels, or object-of-interest tags could further expand the utility of the dataset. We chose to **focus initially on scene graphs because they offer a structured and semantically rich representation** of entities and interactions in the surgical environment. This format aligns well with existing work in surgical scene understanding (4D-OR [2], MM-OR [3]) and enables downstream reasoning tasks in a parsable, explainable manner.
> Importantly, our scene graphs capture all active entities and their interactions at a given timepoint. As such, they can be viewed as a superset of action recognition, from which temporal patterns and higher-level activity descriptions can be derived.
> Regarding natural language and dense captioning, we agree these are highly promising directions. Notably, multiple works have shown how scene graphs can be transformed into descriptive captions, including Transforming Visual Scene Graphs to Image Captions by Yang et al. (ACL 2023\) and Fine-Grained Captioning of Long Videos through Scene Graph Consolidation by Chu et al. (ICML 2025). These methods could be readily applied to our dataset to automatically generate textual descriptions. We will incorporate this discussion in the updated version to clarify how our design supports both current structured tasks and future expansions toward natural-language-based modeling.
>
> **\[2\] Very Targeted Scope: The dataset features only two simulated spine procedures… in a single OR layout.**
>
> We agree and acknowledge that the number of procedures and the single operating room can constrain the dataset’s direct generalizability. However, we believe this reflects the practical challenges of collecting and releasing high-fidelity multimodal surgical data. Capturing and publicly releasing synchronized egocentric and exocentric video, audio, eye gaze, hand pose, ultrasound, and 3D scene information, recorded with non-clinically approved AR glasses, in real surgical settings is logistically, ethically, and legally highly constrained. We view this dataset as a first step toward more comprehensive, diverse benchmarks and actively encourage the entire community to explore expanding into additional procedures and clinical sites.

---

> ### Comment · Area_Chair_5CTf · 2025-08-04
> **Participate in Author-Reviewer Discussion**
>
> Dear VS4V,
>
> Please make sure to read the other reviews and the author response and engage in an open exchange with the authors.
>
> Best,
> AC

---

### Note · Authors · 2025-08-12

We thank all the reviewers for their constructive feedback and for highlighting the novelty of combining multi-member egocentric and exocentric views, the rich multimodality (RGB/RGB-D, gaze, hand pose, audio, ultrasound), the dense frame-level scene-graph annotations, and the usefulness of the benchmark and baseline. In our final version, we clarified the introduction, described synchronization checks, added statistics/flags for scripted edge-case takes, and broadened the discussion of applications, future tasks, and baseline choices. Our consent process and ethics discussion were received positively. Overall, we believe EgoExOR will help pave the way for next-generation ego-exocentric approaches for OR perception and beyond.

---

### Decision · Program_Chairs · 2025-09-18

**Decision:**

Accept (poster)

**Comment:**

This paper proposes a multiview video dataset for surgical activity understanding. Initially this paper received positive to borderline ratings (1 accept, 3 borderline accept). Reviewers appreciated the importance of the problem (VS4V, 3WHi), the multi-person and multi-perspective nature of the dataset (3wHi, LrH6, kA8o), multimodal nature of the dataset (kA8o) the increased amount of data/annotations compared to other surgical datasets (VS4V), the annotation quality/detail (VS4V,LrH6,kA8o), the newly established multimodal baseline (VS4V, LrH6, kA8o) and the clear and thorough description of the collection process (3wHi). However reviewers still had several remaining concerns including the choice of the scene-graph generation task (VS4V), the narrow scope of surgical procedures (VS4V, kA8o), simulated nature of the dataset (kA8o), the potential for misalignment in the camera synchronization (3WHi), limited benchmarking of the dataset (LrH6) and the lack of ablation study to quantify the effect of each modality (3wHi, LrH6).

After the rebuttal reviewers felt the majority of their concerns about had been addressed leading to final scores of 3 accept, 1 borderline accept. The AC agrees with this assessment by the reviewers and that while some weaknesses remain (practical utility) they are outweighed by the strengths and that this paper should be accepted.